# Stereotactic Radiosurgery for Women Older than 65 with Breast Cancer Brain Metastases

**DOI:** 10.3390/cancers16010137

**Published:** 2023-12-27

**Authors:** Rituraj Upadhyay, Brett G. Klamer, Haley K. Perlow, Julia R. White, Jose G. Bazan, Sachin R. Jhawar, Dukagjin M. Blakaj, John C. Grecula, Andrea Arnett, Mariella A. Mestres-Villanueva, Erin H. Healy, Evan M. Thomas, Arnab Chakravarti, Raju R. Raval, Maryam Lustberg, Nicole O. Williams, Joshua D. Palmer, Sasha J. Beyer

**Affiliations:** 1Department of Radiation Oncology, The Ohio State University Comprehensive Cancer Center, Columbus, OH 43210, USA; aiims.rituraj@gmail.com (R.U.); haley.perlow@osumc.edu (H.K.P.); sachin.jhawar@osumc.edu (S.R.J.); dukagjin.blakaj@osumc.edu (D.M.B.); john.grecula@osumc.edu (J.C.G.); arnettandy77@gmail.com (A.A.); mariella.mestres-villanueva@osumc.edu (M.A.M.-V.); evan.thomas@osumc.edu (E.M.T.); arnab.chakravarti@osumc.edu (A.C.); raju.raval@osumc.edu (R.R.R.); joshua.palmer@osumc.edu (J.D.P.); 2Department of Biostatistics, The Ohio State University Comprehensive Cancer Center, Columbus, OH 43210, USA; brett.klamer@osumc.edu; 3Department of Radiation Oncology, The University of Kansas Medical Center, Kansas City, KS 66103, USA; jwhite0581@gmail.com; 4Department of Radiation Oncology, City of Hope Comprehensive Cancer Center, Duarte, CA 91010, USA; jbazan@coh.org; 5Department of Radiation Oncology, University of California, Irvine, CA 92697, USA; ehealy@hs.uci.edu; 6Department of Medical Oncology, Yale Cancer Center, New Haven, CT 06511, USA; maryam.lustberg@yale.edu; 7Department of Medical Oncology, The Ohio State University Comprehensive Cancer Center, Columbus, OH 43210, USA; nicole.williams@osumc.edu

**Keywords:** breast cancer, geriatric cancer, stereotactic radiosurgery, radiation therapy, oligometastases

## Abstract

**Simple Summary:**

There is paucity of data on the optimal management of brain metastases in older women ≥65 years old with breast cancer. In this single-institutional retrospective study, we assessed the survival outcomes and treatment patterns of breast cancer patients ≥65 years old with brain metastases and compared them to younger patients. A total of 112 patients were included. Treatment-related adverse events were similar in both groups. Although older women had a poorer median overall survival compared to younger women (9.5 vs. 14.5 months), survival was similar after adjusting for KPS, extracranial progression, and systemic therapy; and there was no difference in rates of intracranial PFS, neurological deaths, and leptomeningeal disease in the different age groups. This study suggests that age alone may not play an independent role in treatment-selection and that outcomes for breast cancer patients with brain metastases and personalized decision-making including a comprehensive geriatric assessment should be considered.

**Abstract:**

Background: Breast cancer is the second most common cause of brain metastases (BM). Despite increasing incidence of BM in older women, there are limited data on the optimal management of BM in this age group. In this study, we assessed the survival outcomes and treatment patterns of older breast cancer patients ≥65 years old with BM compared to younger patients at our institution. Methods: An IRB-approved single-institutional retrospective review of biopsy-proven breast cancer patients with BM treated with 1- to 5-fraction stereotactic radiation therapy (SRS) from 2015 to 2020 was performed. Primary endpoint was intracranial progression-free survival (PFS) defined as the time interval between the end of SRS to the date of the first CNS progression. Secondary endpoints were overall survival (OS) from the end of SRS and radiation treatment patterns. Kaplan–Meier estimates and Cox proportional hazard regression method were used for survival analyses. Results: A total of 112 metastatic breast cancer patients with BMs were included of which 24 were ≥65 years old and 88 were <65 years old. Median age at RT was 72 years (range 65–84) compared to 52 years (31–64) in younger patients. There were significantly higher number of older women with ER/PR positive disease (75% vs. 49%, *p* = 0.036), while younger patients were more frequently triple negative (32% vs. 12%, *p* = 0.074) and HER2 positive (42% vs. 29%, *p* = 0.3). Treatment-related adverse events were similar in both groups. Overall, 14.3% patients had any grade radiation necrosis (RN) (older vs. young: 8.3% vs. 16%, *p* = 0.5) while 5.4% had grade 3 or higher RN (0% vs. 6.8%, *p* = 0.7). Median OS after RT was poorer in older patients compared to younger patients (9.5 months vs. 14.5 months, *p* = 0.037), while intracranial PFS from RT was similar between the two groups (9.7 months vs. 7.1 months, *p* = 0.580). On univariate analysis, significant predictors of OS were age ≥65 years old (hazard risk, HR = 1.70, *p* = 0.048), KPS ≤ 80 (HR = 2.24, *p* < 0.001), HER2 positive disease (HR = 0.46, *p* < 0.001), isolated CNS metastatic disease (HR = 0.29, *p* < 0.001), number of brain metastases treated with RT (HR = 1.06, *p* = 0.028), and fractionated SRS (HR = 0.53, *p* = 0.013). On multivariable analysis, KPS ≤ 80, HER2 negativity and higher number of brain metastases predicted for poorer survival, while age was not a significant factor for OS after adjusting for other variables. Patients who received systemic therapy after SRS had a significantly improved OS on univariate and multivariable analysis (HR = 0.32, *p* < 0.001). Number of brain metastases treated was the only factor predictive of worse PFS (HR = 1.06, *p* = 0.041), which implies a 6% additive risk of progression for every additional metastasis treated. Conclusions: Although older women had poorer OS than younger women, OS was similar after adjusting for KPS, extracranial progression, and systemic therapy; and there was no difference in rates of intracranial PFS, neurological deaths, and LMD in the different age groups. This study suggests that age alone may not play an independent role in treatment-selection and that outcomes for breast cancer patients with BMs and personalized decision-making including other clinical factors should be considered. Future studies are warranted to assess neurocognitive outcomes and other radiation treatment toxicities in older patients.

## 1. Introduction

Breast cancer is the most common type of cancer seen in women worldwide and the second most common cause of brain metastases [1]. Increasing age is an important risk factor for development of breast cancer with an estimated incidence of 403.8 new patients per 100,000 women ≥ 65 years old versus only 82.2 per 100,000 women aged less than 65 years [2]. With the improved survival of breast cancer patients due to better screening and treatment modalities, the incidence of metastatic breast cancer has increased [3,4]. Although about 5% of women present with brain metastases at diagnosis, 15–30% of women can develop metastases in their disease course [5,6]. The prognosis of patients with brain metastases remains poor, with a median survival of 7.5–7.9 months across studies [7,8].

Systemic therapy is the standard of care treatment for metastatic breast cancer patients [9]. In addition, surgery and radiation therapy have a significant role in management of brain metastases [10,11,12]. With the recent developments in CNS-penetrating targeted therapies, a multi-disciplinary discussion on treatment of brain metastases is critical. However, due to the underrepresentation of the older population in clinical trials, there are limited data on the optimal management of older patients [13,14]. Given their comorbidities, poorer performance status, and lower anticipated survival, whole brain radiation therapy (WBRT) is more often utilized, which can have profound implications on their neurocognitive outcomes and overall quality of life [15,16]. In this study, we describe our treatment patterns and assess the intracranial control and survival outcomes with the use of SRS in older breast cancer patients ≥65 years old with brain metastases as compared to a younger cohort of patients at our institution.

## 2. Methods

### 2.1. Patient Selection

After approval from our Institutional Review Board and data monitoring committee, we identified all patients with biopsy-proven breast cancer and brain metastases treated with stereotactic radiosurgery (SRS) at our institution from January 2015 to December 2021. Study inclusion criteria included: (1) Primary diagnosis of biopsy-proven breast cancer; (2) single or multi-fraction SRS defined as a dose of at least five Gy per fraction given in five fractions or fewer. Patients who received WBRT alone were excluded. Patients were divided in two groups based on age at the time of SRS (<65 years and ≥65 years) and outcomes compared between the two groups.

### 2.2. Radiation Simulation and Treatment Planning

All patients had a thin-slice volumetric post-contrast MRI imaging with at least 1.5 T and were discussed in multidisciplinary conference before proceeding with radiosurgery. Patients were treated on a Varian Edge linear accelerator (Varian, Palo Alto, CA, USA) using noncoplanar volumetric modulated arc therapy (VMAT) or HyperArc©. Patients were simulated supine with a Qfix© Encompass thermoplastic mask (Qfix, Avondale, PA, USA) and treated on a robotic couch with six degrees of freedom, daily kV cone-beam CT image guidance, and surface-guided radiation therapy using the Varian Optical Surface Monitoring System. Gross tumor volume (GTV) was contoured on fused contrast-enhanced MRI images obtained within 2 weeks of SRS treatment. Typical planning target volume (PTV) margin used was 2 mm, though a smaller margin of 1 mm was used for tumors close to the critical structures such as the brainstem and optic structures. Per our linac radiation planning workflow, in case of delays >1-week, either a new planning MRI was obtained or an additional PTV margin of 1–2 mm was added to compensate for the 1–2 weeks in between MRI and treatment [17,18]. The volume for each lesion was determined from physician-defined contours and collected from the treatment planning software. Radiation was planned using a single-isocenter multitarget (SIMT) approach, which permits rapid delivery of focal therapy to multiple brain metastases simultaneously using VMAT [19]. Typical radiation dose prescribed was dependent on fractionation. Selecting single-fraction vs. fractionated SRS was dependent on the number and size of lesions, as well as systemic therapy considerations to minimize delay /interruptions in systemic therapy, while maintaining adequate local control. For patients who received single-fraction SRS, typical prescription dose (marginal dose to the PTV) was 20 Gy for tumors <2 cm in maximal dimension, 18 Gy for 2–3 cm tumors, and 15 Gy for tumors > 3 cm. Patients with overall larger total GTV volume were treated with fractionated SRS over three or five fractions. In addition, if a single-fraction SRS plan could not meet normal brain dose constraint of V12 < 8–10 cc, we considered fractionated SRS. The marginal dose to the PTV was 24 Gy in three fractions (preferred) or 25 Gy in five fractions, with a simultaneous integrated boost to 27 Gy or 30 Gy for three-fraction or five-fraction SRS, respectively. Dose was reduced for an eloquent region like brainstem or optics. There were no dose corrections applied for prior radiation, unless treating the same lesion with repeat SRS.

### 2.3. Follow-Up

Patients were followed every 2–3 months after SRS with contrast-enhanced MRI Brain, including MRI with perfusion. If local or distant treatment failure was diagnosed, patients underwent salvage therapy with surgical resection, additional SRS, or WBRT, based on their disease and clinical performance status. The last clinic visit, imaging, or date of contact was used for censoring patients alive at the time of analysis. Patient-level and tumor-level data for each course were extracted. If a patient underwent multiple courses of SRS, all courses were included separately. Survival outcomes were evaluated based on the first course of SRS. Dosimetric data, including GTV and PTV volumes, were also collected. Follow-up data collected included local treatment failure, intracranial progression, leptomeningeal failure, survival, salvage WBRT, time to salvage WBRT, radiation necrosis (RN), and time to development of RN. Local tumor control was assessed using the Response Assessment in Neuro-Oncology-Brain Metastases guidelines (RANO-BM) [20]. New lesions requiring further courses of RT were classified as intracranial progression. Local treatment failure of the treated lesion(s) was classified separately. RN was defined using surgical pathology or MRI with perfusion and diffusion as available and graded using the National Cancer Institute’s CTCAE version 5.0 (Grade 1: asymptomatic, Grade 2: moderate symptoms, corticosteroids indicated; Grade 3: severe symptoms, medical intervention indicated; Grade 4: life-threatening; urgent intervention; and Grade 5: death). Other adverse events were also graded using the CTCAE version 5.0 [21]. Cognitive outcomes were objectively evaluated using serial NIH Patient-Reported Outcome Measurement Information System (PROMIS)-8 short form scores—using the cognitive function toolkit of the Quality of Life in Neurological Disorders (Neuro-QoL) measurement system [22,23,24]. Follow-up details were extracted from the electronic medical record (EMR) and individual MRI images were reviewed to determine which of the treated metastases were associated with RN or local progression. Data regarding systemic therapy were also collected, including cytotoxic chemotherapy, immunotherapy (any immune checkpoint inhibitors), and/or oral targeted therapies.

### 2.4. Specific Aims/Endpoints

Our primary endpoint was intracranial progression-free survival (PFS) defined as the time interval between the end of SRS to the date of the first CNS progression. Secondary endpoints were overall survival (OS) from the end of SRS and freedom from salvage WBRT.

### 2.5. Statistical Analysis

Summary statistics for patient characteristics are presented as both mean and standard deviation (SD) as well as median and interquartile ranges (IQR). The Pearson chi-squared test was used to assess measures of association in frequency tables. Kaplan–Meier curves were used for survival analysis and the log-rank test was used for intergroup comparisons. OS was defined as the time between initial SRS and death from any cause, with censoring of patients who were lost to follow-up. PFS was defined from the date of SRS to the date of the first intracranial progression or death. Univariate and multivariable logistic regression analyses using Cox proportional hazards models were conducted to evaluate the associations between the clinical or dosimetric factors and survival. Clinically relevant factors including age, performance status, and number of brain metastases, as well as other significant factors on univariate analysis, were included in the multivariable model. Isolated CNS metastasis was not included in the multivariable model due to lack of events in one arm. Logistic regression analyses were summarized using odds ratios and their 95% confidence intervals (CI). A *p*-value of 0.05 or less was considered statistically significant. Statistical tests were based on a two-sided significance level. All statistical analyses were performed using R v4.2.2 (R Core Team) and SPSS v23.0 (IBM Corp, Armonk, NY, USA).

## 3. Results

From January 2015 to December 2021, a total of 112 metastatic breast cancer patients with brain metastases were included of which 24 were ≥65 years old and 88 were <65 yo. They received a total of 43 and 143 courses of linac-based SRS, respectively, to a mean number of 4.3 brain lesions (range 1–22). Patient characteristics are described in detail in Table 1. Median age at RT was 72 years (range 65–84) compared to 52 years (31–64) in younger patients. Median total GTV volume treated was 5.7 cm^3^ (range 0.1–226.7, IQR 1.1–29.2) in older women ≥65 yo. There were significantly higher numbers of older patients with ER/PR positive disease (75% vs. 49%, *p* = 0.036), while younger patients were more frequently triple negative (32% vs. 12%, *p* = 0.074) and HER2 positive (42% vs. 29%, *p* = 0.3). Interestingly, of the 40 patients who underwent surgery for brain metastases, 7 (15.6%) had a change in receptor status compared to the breast biopsy specimen (16.2% in younger patients vs. 12.5% in older). The most common trend was loss of ER positivity (n = 3) followed by gain of ER/PR positivity (n = 2) and gain of HER2 positivity (n = 2). There was also a significant difference among older and young patients in median time from breast cancer diagnosis to development of brain metastases (7 years vs. 4 years, *p* = 0.013); proportion of patients with isolated CNS-only oligometastatic disease (0% vs. 17%, *p* = 0.038); and presence of extracranial disease at SRS (100% vs. 80%, *p* = 0.012). There was no significant difference between the two age groups with respect to ethnicity, Karnofsky performance score (KPS), number of brain lesions treated, number of patients who received prior WBRT, surgery, and systemic therapy (including chemotherapy, targeted therapy, and endocrine therapy) after SRS (Table 1).

Treatment-related adverse events were similar in both groups. Overall, 14.3% patients had any grade RN (older vs. young: 8.3% vs. 16%, *p* = 0.5) while 5.4% had grade 3 or higher RN (0% vs. 6.8%, *p* = 0.7). PROMIS8 scores were available for a limited number of patients (n = 5). Mean scores at baseline, 3 months and 6 months of follow-up were 38, 37.3 and 35, respectively (out of a maximum of 40). When available, serial PROMIS scores were stable for all older patients (n = 3).

Table 2 describes the patterns of failures and survival outcomes. Median OS after RT was poorer in older patients ≥65 years compared to younger patients (9.5 months vs. 14.5 months, *p* = 0.037), while intracranial PFS from RT was similar between the two groups (9.7 months vs. 7.1 months, *p* = 0.580). Figure 1 and Figure 2 demonstrates the corresponding Kaplan–Meier curves of OS and PFS, respectively, for all patients, as well as stratified by age, systemic therapy, and KPS. The rates of freedom from neurological death and leptomeningeal disease (LMD) at 1 year were similar between the two groups (83.3% vs. 87%, *p* = 0.780 and 87.6% vs. 73.3%, *p* = 0.627, respectively).

Table 3 describes the univariate and multivariable relationships of OS. On univariate analysis, significant predictors of OS were age ≥65 year old (hazard risk, HR = 1.70, *p* = 0.048), KPS ≤ 80 (HR = 2.24, *p* < 0.001), HER2 positive disease (HR = 0.46, *p* < 0.001), isolated CNS metastatic disease (HR = 0.29, *p* < 0.001), number of brain metastases treated with RT (HR = 1.06, *p* = 0.028), and fractionated SRS (HR = 0.53, *p* = 0.013). On multivariable analysis, KPS ≤ 80, HER2 negativity, and a higher number of brain metastases continued to predict for poorer survival, while age was not a significant factor for OS after adjusting for KPS, hormone receptor status, extracranial progression, and systemic therapy. Patients who received systemic therapy after SRS had a significantly improved OS on univariate and multivariable analysis (HR = 0.32, *p* < 0.001). Isolated CNS metastases was not included in the multivariable analysis given the low number of events. Specifically, by receptor status, median OS for patients with HER2 positive disease was significantly higher than HER2 negative patients (24.1 months vs. 8.9 months, *p* < 0.001). There was no significant difference by ER positivity (12.4 months vs. 14.6 months, *p* = 0.116) or triple-negative subtype (8.9 months vs. 14.6 months, *p* = 0.167).

On univariate and multivariable analysis of factors affecting the intracranial PFS, the number of brain metastases treated was the only factor predictive of worse PFS (HR = 1.06, *p* = 0.041), implying a 6% additive relative risk of progression for every additional metastasis treated. Table 4 describes the univariate and multivariable relationships of PFS.

## 4. Discussion

Our study represents one of the largest analyses of older women with breast cancer brain metastases who underwent SRS. Our results demonstrate that SRS for older women with breast cancer is safe with low rates of grade 3 or higher adverse events, and intracranial PFS is similar to that achieved in younger patients. Although older women had poorer OS than younger women, this difference was abrogated after adjusting for KPS, extracranial progression, and systemic therapy. This suggests that age alone may not play an independent role in treatment-selection and that outcomes for breast cancer patients with brain metastases and personalized decision-making including other clinical factors should be considered.

### 4.1. Characteristics of Breast Cancer Brain Metastases in the Older Patients

Our patient population was well balanced, with patient tumor characteristics matching the expected differences among the older and younger population. It is well known that older breast cancer patients have higher ER/PR positivity and younger breast cancer patients are more often triple negative [25,26,27], and this was reflected in our cohort of patients as well. About half of breast cancer brain metastases occur in HER2 positive patients, followed by triple negative and then hormone receptor (ER or PR)-positive patients [28]. Interestingly, the median time from primary diagnosis to development of brain metastasis was longer in the older patients compared to the younger patients. This is most likely due to increased incidence of ER+/PR+ in older women. Previous studies have shown that the median time from diagnosis of breast cancer to brain metastasis is significantly shorter for HER2 positive and triple-negative breast cancers than Luminal A and B breast cancers [29,30].

The percentage of patients with CNS-only metastases was significantly higher in the younger cohort (<65 yo) than the older patients. This resulted in the percentage of patients with extracranial disease at the time of SRS being significantly higher in the older population than the younger population. These results are consistent with our unpublished data showing that CNS-only metastases predominantly occur in younger patients with HER2 positive or triple negative breast cancer. All of our patients with CNS-only metastatic disease had HER2 positive breast cancer.

We also looked at the change in receptor status of the brain metastases in patients who had a resection of brain metastases (n = 40) and compared them to the initial breast biopsy specimen. It has been shown that the receptor subtype at the time of recurrence can be discordant from the initial diagnosis [31,32]. Prior studies focused on comparison of receptor status between primary tumor versus metachronous extracranial metastases have reported receptor discordance rates for ER, PR, and HER2 of 10–56%, 25–49%, and 3–16%, respectively [33,34], while discordance for brain metastases is reported to be up to 40% [32,35]. In a meta-analysis, the most common receptor conversions found in breast cancer brain metastases were ER loss (11%), PR loss (15%), and HER2 gain (9.0%). Interestingly, we observed a 15.6% change in receptor status out of 40 patients who underwent a surgery for brain metastases. Our lower rate could be potentially explained by an older patient cohort, as the discordance was slightly higher for younger patients (16.2%) vs. 12.5% in older, though it is difficult to draw any firm conclusions given the limited number of patients that underwent surgery in our cohort. Current guidelines of the American Society of Clinical Oncology advise obtaining tissue diagnosis where feasible, especially in patients who have received prior multiple lines of systemic therapy [36].

### 4.2. Survival Outcomes

We demonstrate similar results to other studies describing outcomes in older breast cancer patients with brain metastasis [37,38]. As may be expected, older patients ≥65 years have poorer OS outcomes than adults <65 years (median OS 10.3 months vs. 14.3 months), according to a post hoc subgroup analysis of a prospective study of SRS in patients with multiple brain metastases [39]. Limited studies have reported outcomes with SRS in breast cancer only patients, especially in the older population. Shen et al. reported the outcomes in 140 patients who underwent craniotomy and reported a median survival of 14.1 months after brain metastasis diagnosis [38]. Their median age was 53 years, and younger age, solitary brain lesion, and HER2 positivity were significant predictors for longer survival. We observed similar results with a median OS of 14.5 months in younger patients and 9.5 months in patients ≥65 years. On multivariable analysis, we noted that KPS > 80, HER2 positivity, and a lower number of brain metastases predicted for longer survival, while age was not a significant factor after adjusting for KPS, hormone receptor status, extracranial progression, and systemic therapy. Molecular subtypes are known to be prognostic for survival and predictive of the response to treatment for brain metastases [40,41,42]. It has been previously shown that although a median survival after recurrence is longest for ER+ luminal A type breast cancers, the survival is shorter in luminal A and triple negative vs. HER2 positive breast cancers when calculated for date of diagnosis of brain metastases [30,43]. We calculated our OS from the date of SRS and hence observed similar findings based on molecular subtypes. This is likely due in part to the recent development of multiple HER2 directed systemic therapies including Trastuzumab, TDM-1, Tucatinib, and Enhertu [44,45].

We observed that patients with isolated CNS-only metastases had an improved OS, which supports the hypothesis that patients with oligometastatic disease have better prognoses. Previous smaller studies have reported that patients presenting with solitary metastasis had a significantly longer median survival than those with multiple lesions [37]. Our study had too few of these patients to confirm this finding.

Interestingly, we observed that OS was improved with the use of fractionated SRS compared to single-fraction SRS. Fractionated SRS leverages fundamental radiobiological differences between brain metastases and surrounding normal brain tissue, which in theory suggests that fractionation can optimize local control while avoiding increased risk for RN [46,47]. Multiple studies have evaluated and compared outcomes with single-fraction SRS and fSRS and observed either no significant difference in local control or necrosis [48,49,50] or slightly lower risks of necrosis [51,52]. Minniti et al. included only patients with brain metastases >2.0 cm and reported a higher local control rate (91% vs. 77%) with fractionated SRS compared with single-fraction SRS in patients with >2 cm brain metastases [53]. Although overall results are controversial, results suggest a strong rationale for using fractionated SRS, particularly when treating larger lesions and postoperative cavities [54].

### 4.3. Radiation-Associated Toxicities

We observed a <15% risk of any RN and ~5% risk of significant, grade 3, or higher RN. This compares favorably to historical outcomes [53,55,56]. There has been some suggestion that older patients may be at higher risk of RN [57], but we did not observe a significant difference in the rates of RN between younger and older population. In a recent retrospective study evaluating outcomes of 119 patients >70 years of age with brain metastases from solid malignancies including breast cancer treated with SRS or WBRT, elderly aged 70–79 years and very elderly ≥80 years did not have a significant difference in acute toxicities rates, suggesting that age alone might not be such a relevant point in therapeutic decision-making [58]. In this study, WBRT was associated with significantly higher rates of any grade 1 to 4 toxicity (OR 7.5) and grade 2 to 4 toxicity (OR 2.8) compared to SRS—this included not only higher rates of cognitive dysfunction, but also increased fatigue, headaches, nausea, and vomiting with WBRT. A recent retrospective cohort study evaluated the risk of symptomatic necrosis with the use of antibody-drug conjugates and concurrent SRS, with a median patient age of 55 years, and observed an overall 24-month cumulative incidence of symptomatic RN at 8.5%, which is similar to our estimate of 10.7% grade 2+ RN [59].

We also evaluated serial PROMIS8 scores for objective assessment of neurocognitive function. While we recognize that number of patients with documented serial PROMIS8 scores was limited, we noted that older patients had a stable score after SRS when available. With improvement in systemic therapies, patients with metastatic breast cancers continue to live longer, and hence patient-reported outcomes and preserving the quality of life become even more relevant. Brain metastasis-related symptoms affect the quality of life (pain, higher neurological functions like language, praxis, reasoning and personality, coordination and strength), especially in the older population. Furthermore, WBRT has shown to have detrimental effects on cognitive functions, especially in older patients > 60 year old [15]. These results suggest that SRS may result in less neurocognitive decline and better quality of life for older breast cancer patients with brain metastases, making its therapeutic ratio potentially greatest in this patient population.

### 4.4. Multi-Disciplinary Treatment of Brain Metastases in Older Women

In this single-institution study, our study showed similar treatment patterns among older patients with breast cancer brain metastases as our younger cohort. Similar rates of local therapies such as SRS, WBRT, and surgical resection were seen among older patients and younger. There was no difference in rates of prior WBRT, median number of lesions treated, radiation dose, salvage WBRT, or surgery. The mean number of SRS courses per patient was also similar among older and younger cohorts.

With recent developments in CNS-penetrating systemic therapies for breast cancer, systemic therapies are important in the multi-disciplinary treatment of breast cancer brain metastases. Recent studies evaluating the use of newer CNS-penetrating agents such as Tucatinib and Trastuzumab Deruxtecan for HER2 positive patients and Sacituzumab Govitecan for triple-negative patients have shown favorable results in treatment of metastatic breast cancer patients [44,45,60]. Eligibility criteria of all of the aforementioned trial did not exclude older patients, and included patients up to 82 years of age [60]. Our data show that there was no significant difference in the number of older and young patients receiving systemic therapy, and patients who received systemic therapy after SRS had a significantly improved OS.

Importantly, our older patient population had a performance status similar to the younger patients we treated. Thus, the patient population in this study may be skewed toward higher performing older patients with fewer comorbidities who are able to tolerate SRS and systemic therapies. This emphasizes the need to consider “the patient as a whole” rather than age alone to guide clinical decision-making in management of metastatic breast cancer patients. Recent ASCO and European taskforce guidelines recommend a comprehensive geriatric assessment (CGA) to identify vulnerabilities or impairments that are not routinely captured in oncology assessments for all patients over 65 years old with cancer [61]. CGA includes assessment of physical and cognitive function, emotional health, comorbid conditions, polypharmacy, nutrition, and social support in older patients. CGA-guided management addresses cancer treatment decision-making as well as impairments through appropriate interventions, counseling, and/or referrals [62]. At our institution, many patients are evaluated by the multidisciplinary “geriatric clinic” that can help perform a detailed assessment of performance status [62]. Our results support that decisions regarding SRS and systemic therapy should be based on comorbidities and overall performance status rather than age alone [63,64].

### 4.5. Study Limitations

Our study is limited by its retrospective design, and therefore inherent biases that affect all retrospective studies, such as the selection bias for who received SRS and a heterogenous patient population. Another major limitation of the study is the imbalance in the number of patients and events in the two patient groups, which may bias the *p*-values. Our patient population was inclusive of patients who had received prior brain radiation to maintain a reasonable sample size. Given the limited sample size and the fact that we did not perform matched cohort comparisons, definitive conclusions cannot be made on the superiority of SRS over WBRT for this population. However, in carefully selected patients, SRS is an appropriate treatment modality for older breast cancer patients ≥65 years old with brain metastases.

## 5. Conclusions

We hereby present our outcomes with the use of SRS for older women ≥65 years old with brain metastases. Although patients ≥65 years of age had poorer OS than younger women, OS was similar after adjusting for KPS, extracranial progression, and systemic therapy; and there was no difference in rates of intracranial PFS, neurological deaths, and LMD in the different age groups. This study suggests that age alone may not play an independent role in treatment-selection and personalized decision-making including other clinical factors is necessary. Future studies are warranted to assess neurocognitive outcomes and other radiation treatment toxicities in older patients.

## Figures and Tables

**Figure 1 cancers-16-00137-f001:**
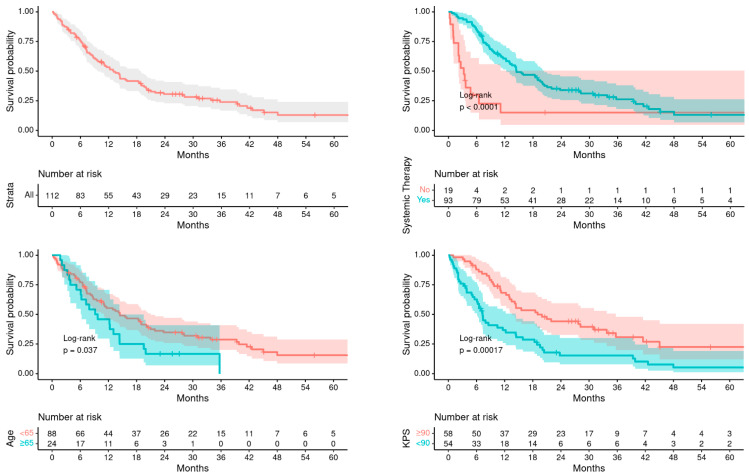
OS for all patients, by age, systemic therapy, and by KPS.

**Figure 2 cancers-16-00137-f002:**
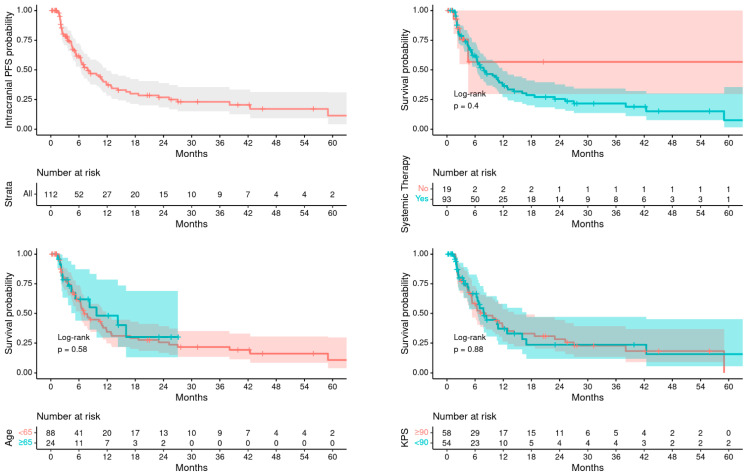
PFS for all patients, by age, systemic therapy, and by KPS.

**Table 1 cancers-16-00137-t001:** Patient characteristics.

	Total N (%)	Age < 65 N (%)	Age ≥ 65 N (%)	*p*-Value
No of patients	112	88	24	
No of SRS courses	186	143	43	
Age at SRS: Median (IQR)	55 (31–84)	52 (44–60)	72 (70–75)	<0.001 *
Breast tumor receptor status ER/PR+ and HER2− ER/PR- and HER2+ ER/PR+ and HER2+ ER/PR- and HER2−				0.034 *
	37 (33%)	23 (26%)	14 (58%)	
	20 (18%)	17 (19%)	3 (12%)	
	24 (21%)	20 (23%)	4 (17%)	
	31 (28%)	28 (33%)	3 (12%)	
ER positive	61 (54%)	43 (49%)	18 (75%)	0.036 *
HER2 positive	44 (39%)	37 (42%)	7 (29%)	0.3
Triple negative	31 (28%)	28 (32%)	3 (12%)	0.074
Ethnicity Caucasians Asians African Americans Others				0.40
	100 (89%)	79 (90%)	21 (88%)	
	2 (2%)	1 (1%)	1 (4%)	
	8 (7%)	7 (8%)	1 (4%)	
	2 (2%)	1 (1%)	1 (4%)	
KPS 90–100 70–80 <70				0.12
	58 (52%)	49 (56%)	9 (38%)	
	50 (44%)	37 (42%)	13 (54%)	
	4 (4%)	2 (2%)	2 (8%)	
Systemic disease Status				
Brain metastases at diagnosis	4 (4%)	3 (3%)	1 (4%)	>0.9
Time from diagnosis to brain metastasis: Median (IQR)	4.0 y (2.0–7.0)	4.0 y (2.0–6.0)	7.0 y (3.0–11.2)	0.013 *
Stable/controlled extracranial disease at SRS	52 (46%)	44 (50%)	8 (33%)	0.2
Isolated CNS metastases	15 (13%)	15 (17%)	0 (0%)	0.038 *
Any Extracranial disease at SRS	94 (84%)	70 (80%)	24 (100%)	0.012 *
Radiation treatment details				
Prior WBRT	19 (17%)	17 (19%)	2 (8%)	0.4
No. of lesions treated: Median (IQR)	3 (1–6)	3 (1–6.5)	3 (1–3)	0.5
Fractionated SRS	87 (78%)	69 (78%)	18 (75%)	0.8
Patients with ≥5 lesions treated	33 (29%)	28 (32%)	5 (21%)	0.4
SRS dose: Median (IQR)	24 Gy (21–25)	24 Gy (21–24)	24 Gy (22–25)	0.2
Salvage WBRT	20 (18%)	17 (19%)	3 (12%)	0.6
Any grade radiation necrosis Grade 1 Grade 2 Grade 3 Grade 4–5	16 (14.3%)	14 (15.9%)	2 (8.3%)	0.5
	4 (3.6%)	2 (2.2%)	1 (4.2%)	
	6 (5.4%)	5 (5.7%)	1 (4.2%)	
	6 (5.4%)	6 (6.8%)	0	
	0	0	0	
Surgery				
Surgery after SRS (Pre-op SRS)	11 (10%)	9 (10%)	2 (8%)	0.8
Surgery before SRS (Post-op SRS)	34 (30%)	28 (32%)	6 (25%)	0.8
Change in receptor status	7 (15.6%)	6 (16.2%)	1 (12.5%)	0.5
Systemic therapy				
Any systemic therapy	93 (83%)	73 (83%)	20 (83%)	>0.9

Abbreviations: SRS: Stereotactic radiosurgery; IQR: Interquartile range; ER: Estrogen receptor; PR: Progesterone receptor; KPS: Karnofsky performance score; WBRT: Whole brain radiotherapy; * denotes variables significantly different between the two age groups.

**Table 2 cancers-16-00137-t002:** Patterns of failure and survival outcomes.

	All Patients N (%)	Age < 65 N (%)	Age ≥ 65 N (%)	*p*-Value
Deaths	85 (76%)	64 (73%)	21 (88%)	0.2
Neurological deaths	14 (12%)	11 (12%)	3 (12%)	>0.9
Intracranial progression	68 (61%)	56 (64%)	12 (50%)	0.2
Extracranial progression	84 (75%)	69 (78%)	15 (62%)	0.12
Leptomeningeal disease (LMD)	17 (15%)	15 (17%)	2 (8%)	0.5
Median OS (95% CI)	13.1 months (9.7–18.9)	14.5 months (11–21)	9.5 months (6.2–14.4)	0.037 *
1-year OS	53%	55%	46%	
Median IC-PFS (95% CI)	7.9 months (6.4–11.9)	7.1 months (6.3–11.8)	9.7 months (5.2–NR)	0.580
1-year IC-PFS	37%	35%	48%	
Freedom from neurological death at 1 year		87%	83.3%	0.780
Freedom from LMD at 1 year		73.3%	87.6%	0.627

Abbreviations: OS: Overall Survival; IC-PFS: Intracranial progression-free survival. * denotes variables significantly different between the two age groups.

**Table 3 cancers-16-00137-t003:** Cox regression univariate and multivariable analysis of OS.

Variable (OS)	Univariate Analysis				Multivariable Analysis		
	N	HR	95% CI	*p*-Value	HR	95% CI	*p*-Value
**Age at SRS (continuous)**	112	1.02	1.00, 1.03	0.084	Not included		
**Age at SRS**				0.048 *			0.055
<65	88	Ref	-		Ref	-	
≥65	24	1.70	1.03, 2.82		1.71	1.01, 2.92	
**ER positive**				0.2			0.7
No	51	Ref	-		-	-	
Yes	61	1.35	0.87, 2.08		1.11	0.69, 1.79	
**HER2 positive**				<0.001 *			0.014 *
No	68	-	-		Ref	-	
Yes	44	0.46	0.29, 0.72		0.54	0.33, 0.89	
**Receptor status**				0.004 *	Not included		
ER-/HER2−	31	Ref	-				
ER+/HER2−	37	1.16	0.68, 1.96				
ER-/HER2+	20	0.38	0.19, 0.78				
ER+/HER2+	24	0.62	0.33, 1.16				
**KPS**				<0.001 *			0.008 *
>80	58	Ref	-		Ref	-	
≤80	54	2.24	1.46, 3.45		1.83	1.17, 2.86	
**Brain metastasis at diagnosis**				>0.9	Not included		
No	108	Ref	-				
Yes	4	1.02	0.37, 2.79				
**Distant extracranial progression**				0.072	Not included		
No	28	Ref	-				
Yes	84	1.65	0.93, 2.95				
**Isolated CNS disease**				<0.001 *	Not included		
No	97	Ref	-				
Yes	15	0.29	0.13, 0.67				
**Previous WBRT**				0.3	Not included		
No	93	Ref	-				
Yes	19	1.38	0.79, 2.42				
**Surgery for metastasis**				0.12	Not included		
No Surgery	67	Ref	-				
Surgery before first SRS	34	0.65	0.40, 1.07				
Surgery after first SRS	11	0.54	0.22, 1.35				
**Fractionated SRS**				0.013 *			0.052 *
No	25	Ref	-		Ref	-	
Yes	87	0.53	0.32, 0.86		0.59	0.36, 0.99	
**Number of brain metastases treated**	111	1.06	1.01, 1.11	0.028 *	1.06	1.01, 1.12	0.023 *
**SRS for >5 lesions**				0.3	Not included		
No	79	Ref	-				
Yes	33	1.26	0.80, 2.01				
**Started systemic therapy after radiation**				<0.001 *			<0.001 *
No	19	Ref	-		Ref	-	
Yes	93	0.32	0.18, 0.57		0.24	0.13, 0.45	

* denotes variables significantly different compared to reference value (Ref).

**Table 4 cancers-16-00137-t004:** Cox regression univariate and multivariable analysis of PFS.

Variable (PFS)	Univariate Analysis				Multivariable Analysis		
	N	HR	95% CI	*p*-Value	HR	95% CI	*p*-Value
**Age at SRS (continuous)**	112	1.00	0.98, 1.02	0.7	Not included		
**Age at SRS**				0.6			0.5
<65	88	Ref	-		Ref	-	
≥65	24	0.84	0.45, 1.57		0.82	0.42, 1.59	
**ER positive**				0.4			0.5
No	51	Ref	-		Ref	-	
Yes	61	1.24	0.77, 2.01		1.21	0.73, 2.01	
**HER2 positive**				0.2			0.5
No	68	Ref	-		Ref	-	
Yes	44	0.71	0.43, 1.16		0.81	0.47, 1.41	
**Receptor status**				0.14	Not included		
ER-/HER2−	31	Ref	-				
ER+/HER2−	37	0.85	0.46, 1.57				
ER-/HER2+	20	0.45	0.22, 0.95				
ER+/HER2+	24	0.92	0.47, 1.77				
**KPS**				0.9			0.6
>80	58	Ref	-		Ref	-	
≤80	54	0.96	0.59, 1.57		0.87	0.52, 1.44	
**Brain metastasis at diagnosis**				0.4	Not included		
No	108	Ref	-				
Yes	4	1.52	0.55, 4.22				
**Distant extracranial progression**				<0.001 *	Not included		
No	28	Ref	-				
Yes	84	4.54	1.95, 10.6				
**Isolated CNS disease**				0.042 *	Not included		
No	97	Ref	-				
Yes	15	0.49	0.24, 1.04				
**Previous WBRT**				0.6	Not included		
No	93	Ref	-				
Yes	19	1.21	0.62, 2.38				
**Surgery for metastasis**				0.10	Not included		
No Surgery	67	Ref	-				
Surgery before first SRS	34	0.54	0.30, 0.97				
Surgery after first SRS	11	0.86	0.40, 1.84				
**Fractionated SRS**				0.14			0.2
No	25	Ref	-		Ref	-	
Yes	87	0.64	0.36, 1.13		0.67	0.36, 1.22	
**Number of brain metastases treated**	111	1.07	1.02, 1.13	0.013 *	1.06	1.01, 1.12	0.041 *
**SRS for >5 lesions**				0.15	Not included		
No	79	Ref	-				
Yes	33	1.48	0.88, 2.49				
**Started systemic therapy after radiation**				0.4			0.7
No	19	Ref	-		Ref	-	
Yes	93	1.55	0.56, 4.30		1.21	0.43, 3.43	

* denotes variables significantly different compared to reference value (Ref).

## Data Availability

Research data are stored in an institutional repository and will be shared upon request to the corresponding author.

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
