# Peer review of "Stereotactic Radiosurgery for Women Older than 65 with Breast Cancer Brain Metastases"

_cancers, 2023, doi:10.3390/cancers16010137_

Round 1

Reviewer 1 Report

Comments and Suggestions for Authors

This is a very interesting retrospective cohort with very reassuring results (even if a limited cohort) about age in BC candidates for RT, as it was not demonstrated to be a significant factor for OS (which was the case for KPS<=80, what indeed underlines the importance of the Comphreensive Geriatric Assessment (CGA) in older patients with breast cancer candidates to stereotactic radiosurgery). Could authors confirm there isn't any CGA data available for this retrospective cohort? I'd suggest to add at on of the discussion paragraphs  one or two sentences about the importance of the CGA upfront to cancer treatment plans (please see ASCO guidelines recently updated).

I'd also suggest to change "elderly" for the more appropriate "older" to avoid ageism interpretations.

Author Response

  1. This is a very interesting retrospective cohort with very reassuring results (even if a limited cohort) about age in BC candidates for RT, as it was not demonstrated to be a significant factor for OS (which was the case for KPS<=80, what indeed underlines the importance of the Comphreensive Geriatric Assessment (CGA) in older patients with breast cancer candidates to stereotactic radiosurgery). Could authors confirm there isn't any CGA data available for this retrospective cohort? I'd suggest to add at on of the discussion paragraphs one or two sentences about the importance of the CGA upfront to cancer treatment plans (please see ASCO guidelines recently updated).

Thank you for your thoughtful review and valuable feedback on our manuscript. Unfortunately, we did not have the Comprehensive Geriatric Assessment data available for this paper. However, we agree that CGA is indeed a very important component in the management of these patients, and we have addressed this in more detail in the discussion. We have added this in the discussion, subsection Multi-Disciplinary Treatment of brain metastases, Page 20 – “Recent ASCO and European taskforce guidelines recommend a comprehensive geriatric assessment (CGA) to identify vulnerabilities or impairments that are not routinely captured in oncology assessments for all patients over 65 years old with cancer [56]. CGA includes assessment of physical and cognitive function, emotional health, comorbid conditions, polypharmacy, nutrition, and social support in older patients. CGA guided management addresses cancer treatment decision-making as well as impairments through appropriate interventions, counseling, and/or referrals [57]”.

  1. I'd also suggest to change "elderly" for the more appropriate "older" to avoid ageism interpretations.

We appreciate your attention to detail and your commitment to improving the clarity and appropriateness of our language. We acknowledge your suggestion to replace the term "elderly" with "older" to prevent potential ageism interpretations. We agree with your concern and recognize the importance of using inclusive and respectful language, in agreement with recent NIH guidelines (https://www.nih.gov/nih-style-guide/age). In light of your suggestion, we have carefully revised the manuscript, replacing instances of "elderly" with "older" throughout the text. We believe that this modification enhances the clarity and appropriateness of our language while maintaining the integrity of the content. We hope that these changes address your concerns, and we remain open to any additional feedback you may have. Thank you again for your time and consideration.

Reviewer 2 Report

Comments and Suggestions for Authors

This is a good work presenting statistical analysis of many crucial variables in two different group of patients with brain metastases from a breast cancer. The statistical analysis is well presented and in line with the results showed but the age based selection of the patients could be a bias in the final analysis of the OS and PFS due to comorbitidies and the longer medical history. This could affect the results showed. 

Overall a good quality paper that I suggest for publication.

Author Response

We appreciate your thoughtful review and thank you for the constructive feedback on our manuscript. We agree that given the retrospective nature of the study, selection bias is a potential problem. We have addressed this in the discussion, page 20, paragraph 2. Thanks a lot for bringing up this important consideration.

Reviewer 3 Report

Comments and Suggestions for Authors

see attached file

Author Response

Specific comments:

  1. Line 64: With regard to prognosis of patients with BM, reference 7 deals with only 42 operated cases. I suggest to add some more representative paper, e.g.: Riecke K et al, Long-term survival of breast cancer patients with brain metastases: subanalysis of the BMBC registry. ESMO Open. 2023 Jun;8(3):101213.; Darlix A et al, Impact of breast cancer molecular subtypes on the  incidence, kinetics and prognosis of central nervous system metastases in a large multicentre real-life cohort. Br J Cancer. 2019 Dec;121(12):991-1000.

We thank you for the thorough review and appreciate your attention to detail and commitment to improve our manuscript. We have modified the statement on the prognosis of these patients and added the aforementioned references. The introduction, para 1, page 4 now reads “The prognosis of patients with brain metastases remains poor, with a median survival of 7.5 – 7.9 months across studies [7,8].

  1. Line 95: About GTV, it would be appropriate to specify the range and median value of GTV volume, which is known to be a critical element for outcome and treatment selection

Thank you for bringing up this important consideration. We have added the median, range and interquartile range of GTV volume in the results, para 1, page 7 – “Median total GTV volume treated was 5.7 cm3 (range 0.1-226.7, IQR 1.1-29.2) in older women ≥65 yo.

  1. Line 96: The authors should discuss the 2-3 weeks delay after MRI, since a shorter timing (either <=48 hours or <= 7 days) is usually recommended, see for example: Kutuk T et al, Impact of MRI timing on tumor volume and anatomic displacement for brain metastases undergoing   stereotactic radiosurgery. Neurooncol Pract. 2021 Jul 28;8(6):674-683. Grishchuk D et al, ISRS Technical Guidelines for Stereotactic Radiosurgery: Treatment of Small Brain Metastases (≤1 cm in Diameter). Pract Radiat Oncol. 2023 May-Jun;13(3):183-194.

Thank you for the comment. 2-3 weeks was the upper limit of obtaining a new volumetric MRI Brain with contrast, but most of our patients had a planning MRI within 1 week prior to treatment. Since a majority (>75%) of our patients were treated with fractionated SRS where we typically add a 1-2 mm PTV margin (depending on tumor location), we believe of 1-2 weeks in between MRI and SRS should not impact outcomes. We have added this in methods, page 5 – “Per our linac radiation planning workflow, in case of delays >1-week, either a new planning MRI was obtained or an additional PTV margin of 1-2 mm was added to compensate for the 1-2 weeks in between MRI and treatment [17,18].”

  1. Line 103-104: Please specify whether dose corrections were applied based on GTV volume or previous WBRT

We have added this information in methods, page 5, which now reads – “Selecting single fraction vs fractionated SRS was dependent on the number and size of lesions, as well as systemic therapy considerations to minimize delay /interruptions in systemic therapy, while maintaining adequate local control. For patients who received single-fraction SRS, typical prescription dose (marginal dose to the PTV) was 20 Gy for tumors <2 cm in maximal dimension, 18 Gy for 2-3 cm tumors and 15 Gy for tumors > 3 cm. Patients with overall larger total GTV volume were treated with fractionated SRS over 3 or 5 fractions. In addition, if a single-fraction SRS plan could not meet normal brain dose constraint of V12 < 8-10 cc, we consider fractionated SRS. The marginal dose to the PTV was 24 Gy in 3 fractions (preferred) or 25 Gy in 5 fractions, with a simultaneous integrated boost to 27 Gy or 30 Gy for 3-fraction or 5-fraction SRS respectively. Dose was reduced for eloquent region like brainstem or optics. There were no dose corrections applied for prior radiation, unless treating the same lesion with repeat SRS.”

  1. Line 103-104: Please explain the criteria you used for choosing fractionated SRS vs single dose

As above, we have added this information in methods, page 5. Over 75% of our patients received fractionated SRS, in 3 fractions. “Selecting the time dose fractionation was dependent on the number and size of lesions, as well as systemic therapy considerations to minimize delay /interruptions in systemic therapy, while maintaining adequate local control. Patients with overall larger total GTV volume were treated with fractionated SRS over 3 or 5 fractions. In addition, if a single-fraction SRS plan could not meet normal brain dose constraint of V12 < 8-10 cc, we consider fractionated SRS.”

  1. Line 228-229: the sentence “Our study represents the largest analysis of breast cancer patients who underwent SRS” does not seem appropriate, unless more explicitly referred only to the elderly subgroup. Indeed, several papers describe larger series, see for example: Sperduto PW et al, The effect of tumor subtype on the time from primary diagnosis to development of brain metastases and survival in patients with breast cancer. J Neurooncol.2013 May;112(3):467-72. Kased N et al, Gamma Knife radiosurgery for brain metastases from primary breast cancer. Int J Radiat Oncol Biol Phys. 2009 Nov 15;75(4):1132-40. Aoyagi K et al, Impact of breast cancer subtype on clinical outcomes after Gamma Knife radiosurgery for brain metastases from breast cancer: a multi-institutional retrospective study (JLGK1702). Breast Cancer Res Treat. 2020 Nov;184(1):149-159.

Thanks a lot for the comment. We have modified this sentence in discussion paragraph 1, page 15 to “Our study represents one of the largest analyses of older women with breast cancer brain metastases who underwent SRS.” We have also included the other references mentioned above throughout the discussion, page 16  - “Previous studies have shown that the median time from diagnosis of breast cancer to brain metastasis is significantly shorter for HER2 positive and triple negative breast cancers than Luminal A and B breast cancers[29,30].” and page 17 - “Molecular subtypes are known to be prognostic for survival and predictive of the response to treatment for brain metastases [40–42].”
